# Hip Surgery in Cerebral Palsy: A Bibliometric Analysis

**DOI:** 10.3390/ijerph20031744

**Published:** 2023-01-18

**Authors:** Norine Ma, Nicholas Sclavos, Kerr Graham, Erich Rutz

**Affiliations:** 1Royal Children’s Hospital, Melbourne 3052, Australia; 2Department of Paediatrics, University of Melbourne, Melbourne 3052, Australia; 3Murdoch Childrens Research Institute, Parkville 3052, Australia

**Keywords:** cerebral palsy, paediatric, hip surgery, hip dislocation, hip subluxation

## Abstract

Hip dislocation in cerebral palsy can lead to pain, pressure sores and difficulty with perineal hygiene. Hip surveillance programs have been implemented to identify patients who might benefit from early intervention and preventive strategies. Surgical techniques used to treat hip dislocation include soft tissue procedures, guided growth, osteotomies and salvage procedures. A search was conducted using Clarivate Web of Science Core Collection on 18 October 2022, to identify all studies of bony or soft tissue surgery for hip pathology in children with cerebral palsy. Fifty-nine original studies and reviews with at least 20 citations were included in this bibliometric analysis. We found that there has been an increase in studies over the decades, with the most studies being published in the *Journal of Pediatric Orthopaedics*. The United States of America was the most productive country, with Boston Children’s Hospital and Harvard University publishing the most articles. The Methodological Index for Non-randomized Studies (MINORS) scoring system was used to analyse the methodological quality of included cohort studies, with the median score being 11 out of 18; many studies had no prospective calculation of study size and lacked control groups. Overall, the literature on this topic appears to be preferentially published in the *Journal of Pediatric Orthopaedics*, and influential papers by Hagglund 2005 and 2014 continue to be highly cited.

## 1. Introduction

Cerebral palsy (CP) results from non-progressive injury to the brain during fetal or infantile development and can lead to neurological deficits that impact mobility [1]. Hip development and function can be impaired in children with cerebral palsy and can result in contractures and fixed deformities around the hip, hip dysplasia, subluxation, dislocation and rotational deformities. The traditional view is that hip displacement (HD) in CP is caused by hypertonia and muscle imbalance of the pelvi-femoral muscles, increasing joint contact forces and instigating increased force vectors in a direction favourable to displacement and dislocation [2]. A contrary view states that the negative features of the upper motor neurone (UMN) syndrome may be more important, resulting in coxa valga, persistent or increased femoral neck anteversion (FNA) and HD [3]. Hip displacement can lead to: pain; pressure sores; difficulty with perineal hygiene; loss of function, including the ability to sit or stand; impaired health related quality of life (HRQoL); and increased burden of care. Loss of hip joint congruence is particularly prevalent in children with severe bilateral cerebral palsy who are non-ambulatory: Gross Motor Function Classification System (GMFCS) levels IV and V [3,4,5]. Concurrent musculoskeletal deformities may include pelvic obliquity and scoliosis, which also need to be considered in overall management strategies [6,7]. As a result, several orthopaedic surgery techniques have been developed over the years to manage hip pathology ranging from preventive measures such as abduction bracing, abduction bracing combined with injections of Botulinum Toxin A, soft tissue procedures (adductor and psoas lengthening), reconstructive osteotomies of the femur and pelvis, and salvage procedures resulting in loss of the hip joint [2,8,9].

Early intervention has shown promise in preventing progression of hip pathology, and avoiding the need for more significant operations [10]. Hip surveillance (HS) protocols have been developed and are used in many countries to aid in the early identification of patients who would benefit from preventive or early reconstructive strategies [11,12,13]. The features common to most HS programs are standardised history taking and physical examination, combined with monitoring of hip development using radiographs taken at regular intervals. The frequency of radiographs aims to strike a balance between early detection of HD and avoidance of excessive ionising radiation. In most HS programs the frequency of radiographs is dictated by age and GMFCS level, with more frequent surveillance recommended for patients with higher GMFCS levels due to their higher risk for progressively worsening hip subluxation [14]. In non-ambulatory patients, surveillance frequency is also determined by concerns with pelvic obliquity, leg-length discrepancy and worsening gait, often continuing until at least skeletal maturity.

The radiographical features commonly interpreted to guide treatment plans include Reimers’ Migration Percentage (MP), which is measured on anteroposterior hip plain radiographs and based on the proportion of the femoral head that is covered by the acetabulum [15]. Many other radiographical features have been investigated to determine their relevance to the assessment of hip stability, including femoral head shape and head shaft angle, as concurrent femoral head and neck deformities have been associated with hip subluxation [16,17]. The Melbourne Cerebral Palsy Hip Classification Score (MCPHCS) also considers radiographical features in order to categorise the hip status of patients [18]. This includes evaluation of the migration percentage, Shenton’s arch, femoral head shape, acetabular shape and pelvic obliquity. The MCPHCS provides a system for categorisation of hip pathology in patients to allow ease of communication between health professionals and in research settings.

Surgery can be offered to patients where hip subluxation results in difficulties with nursing care or to prevent progressively worsening incongruency of the hip joint. Soft tissue procedures including adductor or iliopsoas tenotomy are effective in preventing the advancement of hip subluxation to hip dislocation but the treatment effect is small and usually short lived [19]. Bony procedures have also been successful, such as femoral varus-derotation osteotomy (VDRO) to reposition the femoral head relative to the acetabulum [20,21]. Combined femoral and pelvic osteotomy is often reported to be more effective and long lasting than VDRO alone. Salvage procedures such as proximal femoral resection arthroplasty are preferably avoided as these have a high rate of failure and some patients ultimately require revision surgery due to an unsatisfactory primary procedure [22].

The topic of surgery for hip pathology in patients with cerebral palsy has been widely studied, and there are several key articles that have moulded current management practices. This study is a bibliometric analysis of studies relating to surgery for hip pathology in cerebral palsy including preventative, reconstructive and salvage procedures for hip subluxation or displacement. It aims to identify and explore publication trends over the years, including productivity of countries, institutions, authors and journals in the field. It also aims to visualise the connectivity between authors, citations and keywords through network mapping.

## 2. Materials and Methods

The Clarivate Web of Science Core Collection database was searched for abstract fields from 1900 to 18 October 2022 using the keywords “cerebral palsy” and “surg*”. These main keywords were used in order to capture as many relevant articles as possible in the field of surgery in cerebral palsy, with articles relating to surgery for hip pathology manually extracted from the subsequent list of papers. Articles were screened using the inclusion criteria in Table 1. Only studies of surgical management of hip pathology were included, with the exclusion of selective dorsal rhizotomy. All included studies were either original studies or reviews, and a minimum of 20 citations was chosen as the threshold for inclusion.

Data collected included authors, institution, journal, year of publication and number of citations. Clarivate Journal Citation Reports was used to determine the 2021 impact factor for included studies, except for *The Journal of Bone and Joint Surgery—British Volume*, where the 2014 impact factor was used due to discontinuation of the journal.

Publications were grouped by decade of publication, with the average number of citations calculated per article per decade. The average number of citations corrected for years since publication was also calculated for each decade time period. The most productive countries, institutions, authors and journals were determined by number of publications. Average citations per publication were calculated for the most productive authors, as well as for each journal.

There were 48 cohort studies which were assessed for methodological quality using the Methodological Index for Non-randomized Studies (MINORS) scoring system [23]. This provides a score out of 18 for cohort studies, based on inclusion criteria, prospective or retrospective collection of data, appropriate and unbiased endpoints, follow-up and suitable statistical analysis [19]. A further 5 studies were comparative studies and so received a MINORS score out of 24, to include three additional criteria for the quality of the comparator group. There was one randomised controlled trial and five review articles which were not suitable for quality assessment using MINORS.

VOSviewer version 1.6.18 was used to construct network visualisations for bibliometric analysis using the fractional counting method. Network visualisations were constructed using data extracted from the Clarivate Web of Science Core Collection. Networks were created to represent co-citation by authors, bibliographic coupling by authors and co-occurrence of keywords.

## 3. Results

The search yielded 3139 articles in total, with 59 articles meeting the inclusion criteria (Appendix A). Of these articles, 54 were original studies and 5 were review articles.

The articles included were published between 1975 and 2016 inclusive, with more articles in the latter years (Figure 1). More recent articles from 2017 onwards were excluded, however, due to the threshold of 20 citations chosen for inclusion; it was expected that these newer articles would not have accumulated enough citations to be included. A full list of eligible studies is presented in the Appendix A, including a separate table for articles between 2017 and 18 October 2022 that had between 10 and 20 citations, and so did not meet the inclusion criteria for this analysis.

The trend for number of publications per year opposes the average number of citations per article, which decreased in later decades (Figure 2) from 68 average citations per article from 1970 to 1979, to 41 average citations per article from 2010 to 2019. This is to be expected, as more recent articles would have had less time to develop a large number of citations. Figure 2B shows a further analysis of average citations, taking into account the number of years since each article was published. The figure shows that articles published in more recent decades have a higher number of citations per year. Articles published from 1970 to 1979 had an average of 1.45 citations per year across the lifetime of the articles. Articles published from 2010 to 2019 had an average of 4.77 citations per year.

The most highly cited article was “Prevention of dislocation of the hip in children with cerebral palsy—The first ten years of a population-based prevention programme” by Hagglund in 2005, with 181 citations. The United States of America (USA) published the most articles (36 articles) followed by Australia (5 articles) (Table 2). The most productive institutions were Boston Children’s Hospital (USA) and Harvard University (USA), both with 5 publications (Table 3). Two authors, Dabney (USA) and Miller (USA), published the greatest number of articles (4 articles each) (Table 4).

*The Journal of Pediatric Orthopaedics* published the most articles (22 articles) followed by *The Journal of Bone and Joint Surgery—American Volume* (9 articles) (Table 5). Most articles were published in a journal with an impact factor between 2 and 3 (noting that *The Journal of Pediatric Orthopaedics* has an impact factor of 2.537 and makes up 22 of the 32 articles within this range.

The median MINORS scores for the 48 cohort studies was 11 out of 18, indicating a fair quality of evidence (Figure 3). A MINORS score of <10 can be interpreted to indicate low quality of evidence, 10 ≤ MINORS score < 14 to indicate fair quality of evidence and MINORS score > 15 to indicate good quality of evidence. Most of the reviewed studies were in the “fair quality” category.

The co-citation of authors was visualised (Figure 4) showing authors that are cited together in at least 5 documents. Connectiveness is increased when there are more documents where authors are cited together. This shows four groups of authors who are cited together frequently.

Bibliographic coupling by authors was visualised (Figure 5) where connectiveness was determined by the overlap of reference lists between documents by these authors. The strength of connection is increased if there is a third author that is commonly cited by two authors. Only those with at least two common authors cited were included. This visualisation shows groups of authors who commonly cite the same authors.

The co-occurrence of keywords was visualised (Figure 6) to demonstrate keywords that appear together in at least two papers. Connectiveness is determined by the number of papers in which the keywords appear together. This demonstrates the different areas in hip surgery that are studied, including classification, osteotomy, arthroplasty and adductor release.

## 4. Discussion

There has been a rapid increase in the output of research publications in the medical field over recent years, with a similar pattern observed in the field of paediatric hip surgery in children with cerebral palsy. As would be expected, this was associated with a decrease in the total citation numbers of more recently published articles. However, the number of citations per year of an article’s lifetime was higher in more recently published articles. It can be seen that landmark articles such as Hagglund 2005 [24] and Hagglund 2014 [25] continue to be widely cited and form an important foundation for ongoing research in the area. Hagglund’s studies on a Swedish hip surveillance programme are population-based, included large sample sizes (over 250 patients) and relatively long follow-up periods (10 and 20 years), which allow the findings to continue to be relevant and highly cited within the field.

The most productive journal was The Journal of Pediatric Orthopaedics. This may be attributable to many factors including the popularity of the journal, or a preference by authors in this field to aim for publication in this journal instead of others. It could also be explained by the threshold of 20 citations chosen in this study, which may preclude the inclusion of less well-known journals, in which articles may not easily be read or become known. Conversely, the over-representation of this journal may give an indication of the perceived quality of the journal and its publications compared to other journals when authors are choosing to cite references. Nonetheless, The Journal of Pediatric Orthopaedics did not represent the highest impact factor nor the highest average citations per article. There did not appear to be a correlation between the impact factor, number of articles and average number of citations per article, at least in the subset of papers included in this study. The journal with the highest impact factor (6.558) was The Journal of Bone and Joint Surgery, American Volume. However, it has a similar average number of citations per article when compared to The Journal of Pediatric Orthopaedics (impact factor 2.537). The journal with the highest average number of citations per article (105) was The Journal of Bone and Joint Surgery—British Volume, however its impact factor of 3.309 was the median of the group of journals included. While the quality of journals is often perceived to be based on their impact factors, some research areas may have a tendency to gravitate towards particular journals, which may in turn be informally accepted as the leading publisher on the topic of interest.

The most productive country, journal, and authors were all from the USA. All of the top eight most productive institutions were in the USA with the exception of the Royal Children’s Hospital, Melbourne, Australia. The top eight most productive authors were determined by the number of publications per author. However, when considering the average number of citations per article, as a pseudo-measure for article impact, Graham HK was the most impactful author. When looking at the citation count, Hagglund and co-authors had the highest number of citations across the two papers that were included in this analysis.

While looking at citation count can help to determine the overall impact of countries, authors and journals, it is only one measure to determine the impact on the scientific literature base. It should be noted that there is no easy way to determine whether these citations are positive or negative, as studies may commonly be cited to show a shift in thinking or further research may have superseded the conclusions made in some older studies. Logically, the impact of scientific literature would be a function of its reach and the general acceptance of its overall findings and conclusion. The citation count of an article does not fully encompass all the factors affecting an article’s impact. The h-index has been proposed by Hirsch as a way to evaluate the impact and relevance of an individual’s research output [26]. This is a function of the number of papers an author has published, and the number of citations for each paper. However, this would mean that an author who has published fewer but more frequently cited articles would have a lower h-index than an author who has published a large volume of articles with fewer citations per article. It is also difficult to compare authors across different research fields, as those who publish in extensively researched areas would naturally receive many more citations than researchers in more niche fields. It is also important to note that publication output by authors does not follow a linear pattern, and indexes such as the h-index do not fully take into account the fluctuations in productivity over an author’s career. Another index proposed by Hirsch to account for this is the m-quotient, which helps to compare authors across different career points. The m-quotient takes the h-index and divides it by the number of years since first publication. This balances out the assessment of successful early-career researchers who may have fewer but frequently citated publications, and late-career researchers who may have numerous publications but fewer citations per publication.

There remains to be developed a more comprehensive means of determining the impact of a researcher, journal or article, which can be used as a comparison within or between fields. Many factors would need to be considered, including: the number and pattern of citations over the life of an article; the publisher and reach of the journal; positive and negative citations; the quality of studies, such as level of evidence; and the career stage of the researcher. In addition, it should be noted that the number of citations can be affected by many sources of bias such as self-citation, a tendency to cite based on publisher or country of the article, avoiding citing competitor research groups, and the exclusion of non-English-language studies in many articles. Articles also tend to accumulate citations in a non-linear pattern, meaning the total citation count may not necessarily provide an indication for past or current interest in the article. Likewise, the productivity of authors also fluctuate through the lifetime of their career, with authorship position also a consideration for the contribution of an author to their published research output.

The MINORS scores were useful to indicate how future study quality can be improved. The most frequent study limitations were the lack of a control group, no prospective sample size calculation and unblinded study endpoints. These are limitations that are not uncommon in surgical studies due to the difficulty of having an appropriate comparator or control group. Additionally, sample sizes in surgical studies relating to cerebral palsy are often small due to the frequency with which these surgical procedures are conducted within an institution. The recent introduction of guided growth could provide a minimally invasive comparator for other preventive options such as soft tissue releases, which would allow the development of more rigorous study designs in this field.

There were some limitations in this bibliometric analysis. In this study, a minimum inclusion criteria of 20 citations was chosen in order to ensure that the studies included had been adequately cited. This resulted in newer publications being excluded as they inherently have had less time to accumulate citations. Interestingly, articles published in more recent decades have, on average, a higher number of citations per year since publication. This could be an indication of the life cycle of an article; there may be a peak in the number of citations an article receives when it is recently published due to the new evidence contributed to the field, followed by a downtrend. The downtrend may be due to articles falling out of favour or popularity, or older articles being superseded by more recent evidence. It could also indicate fluctuating interest in a field of study leading to a drop in citations per year for an article as time passes.

Another limitation is the variability of impact factors each year. In this study the 2021 impact factor was used as this was the most recent one available at the time of analysis. This would not have been the impact factor for the journal in the year that the paper was published. However, it should be noted that many journals’ impact factors increase over time, so using the impact factor for the year of publication of each paper would introduce bias towards more recently published papers.

## 5. Conclusions

Surgery for hip pathology in children with cerebral palsy continues to be widely studied, with interest from several research groups around the world. The USA provides the greatest number of articles in the field, with The Journal of Pediatric Orthopaedics most highly involved. There has been an increase in the research output in this field, in particular in the last two decades, however landmark papers such as Hagglund 2005 continue to be highly cited. There is room for improvement in study-design quality for future studies, in particular the use of prospective study designs, larger sample sizes and a consideration of appropriate comparison groups.

## Figures and Tables

**Figure 1 ijerph-20-01744-f001:**
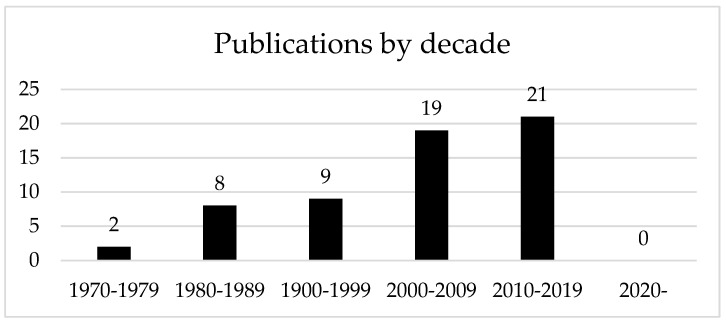
Publications by decade.

**Figure 2 ijerph-20-01744-f002:**
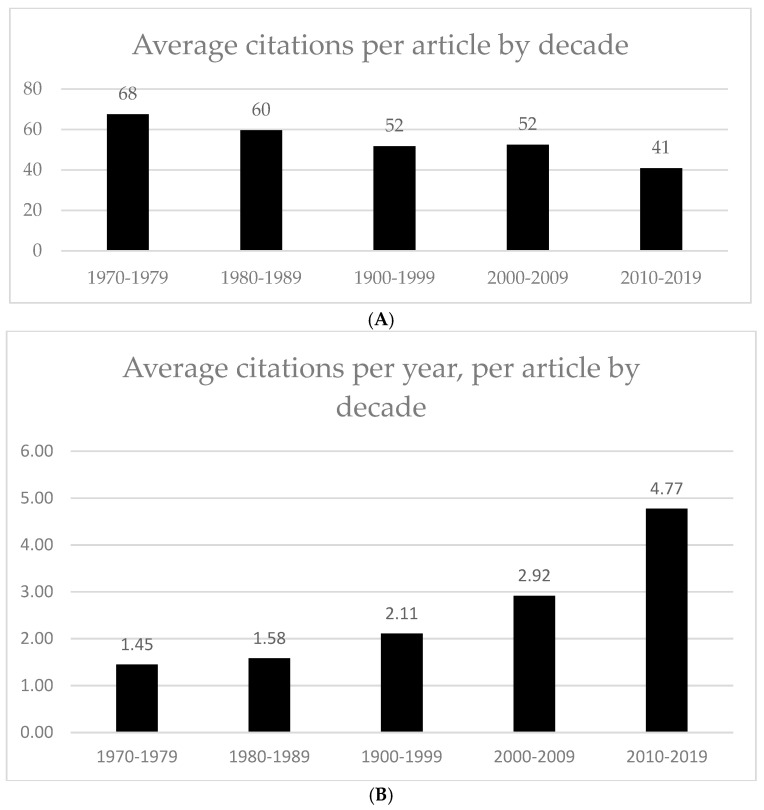
(**A**) Average citations per article by decade. (**B**) Average citations per year, per article by decade. Calculated by number of citations, divided by number of years since publications (using 2022 as the index year). This was then averaged over the articles published within each decade as shown.

**Figure 3 ijerph-20-01744-f003:**
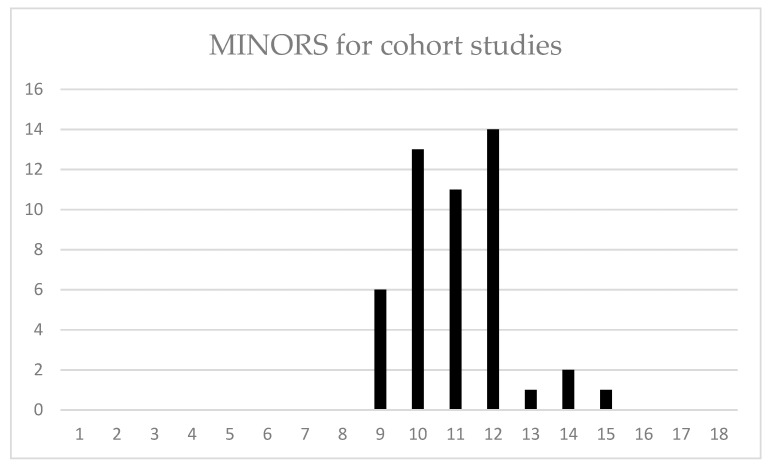
MINORS scores for cohort studies.

**Figure 4 ijerph-20-01744-f004:**
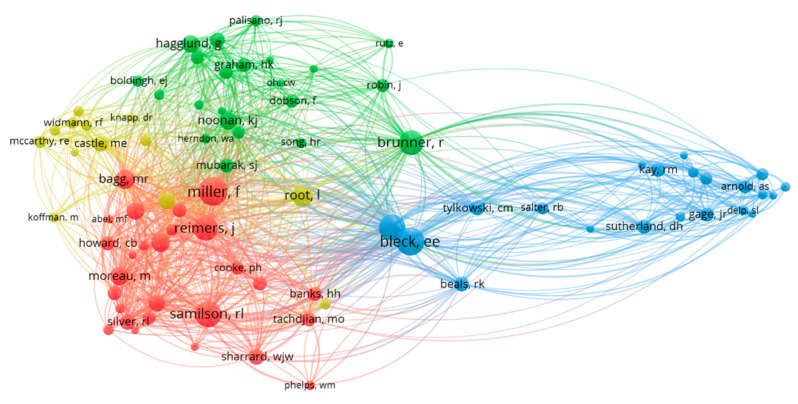
Co-citation by authors.

**Figure 5 ijerph-20-01744-f005:**
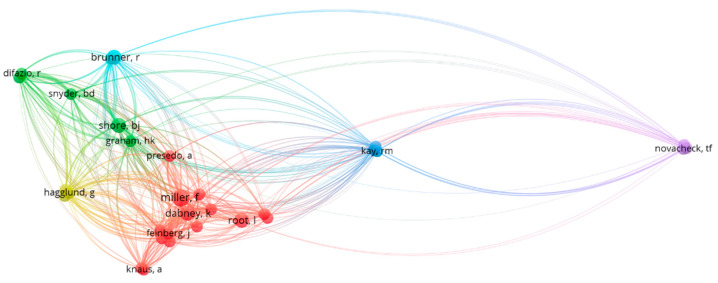
Bibliographic coupling by authors.

**Figure 6 ijerph-20-01744-f006:**
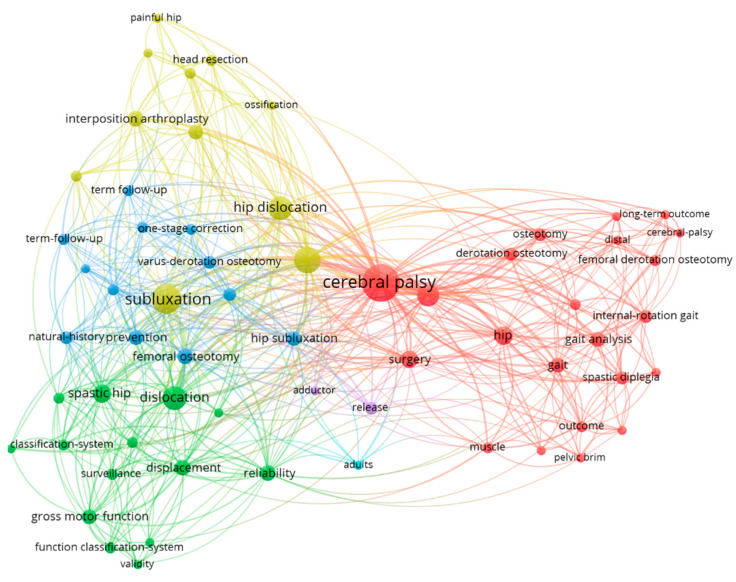
Co-occurrence of keywords.

**Table 1 ijerph-20-01744-t001:** Inclusion and exclusion criteria.

Inclusion Criteria	Exclusion Criteria
Studies including children with cerebral palsy	Selective dorsal rhizotomy
Bony and/or soft tissue surgery for hip pathology	Physical therapy or physiotherapy alone as intervention
Hip dislocation, subluxation, dysplasia, rotation or pain	Botulinum toxin A injection for hip function alone as intervention
Minimum of 20 citations (inclusive)	
Original studies or reviews	

**Table 2 ijerph-20-01744-t002:** Top 5 most productive countries.

Country	Number of Articles
USA	36
Australia	5
England	3
South Korea	3
Switzerland	3

**Table 3 ijerph-20-01744-t003:** Top 8 most productive institutions.

Institution	Number of Articles
Boston Children’s Hospital (USA)	5
Harvard University (USA)	5
Nemours Alfred I Dupont Hospital For Children (USA)	4
Royal Children’s Hospital Melbourne (Australia)	4
Children’s Hospital Los Angeles (USA)	3
Gillette Children’s Specialty Healthcare (USA)	3
University of Minnesota System (USA)	3
University of Minnesota Twin Cities (USA)	3

**Table 4 ijerph-20-01744-t004:** Top 8 most productive authors.

Author	Number of Articles	Number of Citations	Average Citations Per Article
Dabney KW	4	247	62
Miller F	4	250	63
Brunner R	3	131	44
Graham HK	3	228	76
Novacheck TF	3	137	46
Root L	3	150	50
Schwartz MH	3	137	46
Shore BJ	3	134	45

**Table 5 ijerph-20-01744-t005:** Journals.

Journal	Number of Articles	Number of Citations	Average Citations Per Article	Impact Factor
*Journal of Pediatric Orthopaedics* (USA)	22	1036	47	2.537
*Journal of Bone and Joint Surgery, American Volume* (USA)	9	479	53	6.558
*Gait & Posture* (Netherlands)	5	203	41	2.746
*Journal of Pediatric Orthopaedics, Part B* (USA)	5	167	33	1.473
*Journal of Bone and Joint Surgery—British Volume* (UK)	4	420	105	3.309
*Developmental Medicine and Child Neurology* (USA)	4	184	46	4.864
*Orthopedic clinics of North America* (USA)	3	146	49	2.771
*Bone & Joint Journal* (UK) *	2	167	84	5.385
*Acta Orthopaedica* (Scandinavia)	2	51	26	3.925
*Journal of Children’s Orthopaedics* (Germany)	2	47	24	2.537
*Clinical Orthopaedics and Related Research* (USA)	1	32	32	4.755

* Formerly Journal of Bone and Joint Surgery—British Volume.

## Data Availability

The data presented in this study are available in the Appendix A.

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
