# Peer review of "Hip Surgery in Cerebral Palsy: A Bibliometric Analysis"

_ijerph, 2023, doi:10.3390/ijerph20031744_

Round 1
Reviewer 1 Report
The manuscript titled ” Hip surgery in Cerebral Palsy: A Bibliometric Analysis” contains the original information that progresses knowledge about the bibliometric analysis and the impact of published research on the current status of pediatric orthopedic literature of hip pathology observed in cerebral palsy children. This is a conceptual well-established study it attempts to define the current trends and goals of future publications in this field.
The topic established the main goal of the paper is compatible with the scope of the journal. The aim of the study, described in the last section of the Introduction part, should be more clear. It is difficult to focus the study on the improvement of future study quality with the exclusion of all publications from the last three years (2000-2022). It would be interesting to explain the reasons for that situation. The subject of evaluation is presented with a high level of accuracy and balance between the obtained results and suspected limitations. In the scope of the current literature, the presented results include practical aspects for all researchers and clinicians dealing with children with cerebral palsy about how to use and how to search the literature to get the answer to clinical or research questions about hip surgery in cerebral palsy.
However, there are a few limitations that should be explained:
1. the reasons for taking under consideration just only two keywords "cerebral palsy" and "surg" with the exclusion of "hip" or "hip joint"?
2. the reasons for the lack of publications from the last 3 years cited in this article - probably it is because of too short follow-up time. The research should be time limited to the period from 1900 to Dec 31, 2019, not to 18, October 2022
Reviewer 2 Report
This is a very well-written paper. My comments and suggestions are below which are all easily remedied with a revision or justification.
Why was the minimum of 20 citations chosen as the threshold for inclusion? Is this standard practice for a bibliometric analysis? This may bias the analysis towards exclusion of more recent publications that have not had the time to build the absolute number of citations to 20, especially considering the end date for article examination was only a few months ago. Further, the topic is a relatively niche area, so fewer citations is expected, especially if they were published recently. The potential for this bias is supported by the lack of studies after 2016.
I have 2 suggestions based on what the authors might think is most appropriate for their intended goals. One suggestion is to drop the criterion for 20 minimum citations to avoid this bias completely, and then update the analysis and text. This is ideal to avoid the bias that directly impacts this work’s interpretation. If not possible, the other suggestion is to do a few things. First, provide a brief justification as to why a 20 citation minimum was used in the Methods section for context (lines 86-87). Second, add a new supplementary table 2 that includes the papers that were excluded purely for this criterion, similar to the style of Supplementary Table 1 of the included studies. Third, add a limitation regarding that this analysis is biased away from more recent papers, as they have not had enough time to develop the absolute minimum of 20 citations in a relatively niche field. This at least increases transparency.
Does Figure 2 account for the time since publication? Since a conclusion is regarding productivity, it seems important to account for time. Perhaps a Figure 2B that divides each article’s citation # by the number of years from 2022 since publications (2022 minus the year the article was published). This does not fully equalize the difference in time, considering linear vs. polynomial citation trajectories, but helps account for it somewhat.
Table 5: Is the impact factor as of October 18, 2022? Or some sort of average based on the year each article was published? If the former, this may be important to briefly note in the Discussion; i.e., that impact factors change over time especially over the decades examined in this study.
I recommend adding relative metrics to the tables and figures where relevant, like discussed for Figure 2 above. Accounting for time in bibliometric analyses is important, but challenging as it does not fully account for polynomial trajectories. Nevertheless, accounting for time provides a relative assessment that complements that absolute assessment. For example, the authors discuss the H-index. The M-index has been proposed, which divides an author’s H-index by the number of years active (I believe since the year of their first publication?), and helps to compare productivity across various career stages- but again, not perfect.
